# Psychological capital and mental health problems among the unemployed in the post-COVID-19 era: Self- esteem as a moderator

**Cao Tran Thanh Trung**[1,2]**, Nguyen Tan Dat**[2,3]*****, Choon Jin Teh**[1]**, Poh Kiong Tee**[4]

**1** School of Marketing and Management, Asia Pacific University of Technology & Innovation, Kuala Lumpur, Malaysia, **2** Lumos Psychotherapy and Counseling Center, Ho Chi Minh City, Vietnam, **3** Department of Psychiatry, Hokkaido University Graduate School of Medicine, Sapporo, Japan, **4** School of Management and Business, MILA University, Nilai, Malaysia

* nguyentandat23496@gmail.com

## Abstract

### Introduction

Since the coronavirus pandemic outbreak, unemployment has become a widespread phenomenon in society, with notable consequences including the emergence of mental health problems. This study examined the influence of psychological capital on mental health issues among unemployed people in Vietnam as well as the moderating role of self-esteem in this relation.

### Methods

The study adopted a quantitative, cross-sectional approach with 468 recently unemployed office workers in 2023. The participants ranged from ages 22 to 49 years and worked in different fields including technology, finance, consumer services, and infrastructure services.

### Result

The results indicated severe to extremely severe mental health symptoms, with 54.5%, 50.9%, and 38.9% of respondents reporting symptoms of depression, anxiety, and stress, respectively. Our results have shown that psychological capital can significantly predict mental health issues including depression, anxiety, stress, and suicidal ideation. Self-esteem was found to play a moderating role in the relation between psychological capital and stress, depression, and suicidal ideation but not anxiety.

### Conclusions

Businesses or government agencies should provide mental health support for unemployed workers. Employees should realize that they must improve their psychological capital and self-esteem to propose mental well-being in the post-pandemic period.

**Data availability statement:** The dataset for the current study is provided in the Supporting Information.

**Funding:** The author(s) received no specific funding for this work.

**Competing interests:** The authors have declared that no competing interests exist.

# 1. Introduction

The impact of the coronavirus disease 2019 (COVID-19) pandemic is profound, affecting the social, political, economic, and healthcare sectors globally. In the post-COVID-19 era, the world faced a socioeconomic crisis marked by significant events, including bankruptcies among major banks, rising inflation, and recessions [1,2]. As a result of this crisis, companies are forcing them to implement mass layoffs or reduce their workforce to maintain operating costs [3,4]. In 2023, more than 260,000 employees from over 1,100 technology companies experienced layoffs, an increase of nearly 58% compared with 2022 [5]. Although the global economy has been affected by the COVID-19 pandemic, Vietnam continues to be one of the few developing countries attracting strong Foreign Direct Investment (FDI) flows in the world [6]. However, over-reliance on FDI and exports has made the country's economic performance very vulnerable to external shocks [7]. This has led multinational corporations with branches in Vietnam to drastically cut headcount. Specifically, the wave of layoffs in this country started in early 2023 in diverse fields such as manufacturing, finance, real estate, and technology [8,9]. Notably, the Ministry of Labor reported that 280,000 Vietnamese workers lost their jobs in the first 6 months of the year [10].

Layoffs can be a traumatic event with negative impacts on workers' mental health. Specifically, high layoff rates have been linked to serious mental health issues such as major depressive disorders, dysthymia [11], anxiety [12], posttraumatic stress disorder [13], and even suicide [14]. Accordingly, more studies must be conducted on protective factors against mental health issues among unemployed workers.

Personal resources can act as a buffer against psychological issues in unemployed individuals [15], a concept supported by the JD-R model developed by Bakker and Demerouti. The JD-R model, applied across various work environments, suggests that stress arises when there's an imbalance between the demands placed on individuals and the resources they possess to meet those demands, with stronger personal resources offering greater resistance to mental health symptoms [16].

Psychological capital (PsyCap) is considered a personal resource that is frequently observed among employees in various business environments [17,18]. PsyCap is defined as "a state of positive psychological development of an individual", and characterized by four components: hope, resilience, optimism, and self-efficacy [17]. Based on positive psychology constructs and empirical research, PsyCap refers to an individual's ability to manage and react to difficult situations, facilitating the development of healthy individuals [17]. Each component of PsyCap plays a vital role in reducing mental health problems. As a synergistic effect, hope and optimism work together to create a positive mindset, while self-efficacy and resilience strengthen an individual's belief in their ability to overcome hindrances [17,19]. In other words, PsyCap, as highlighted by Luthans and Youssef-Morgan [17], is instrumental in preventing mental health issues by fostering adaptive coping strategies and empowering individuals to confront life's challenges with resilience and optimism. Although it is a relatively new concept, PsyCap has had a significant positive impact on many fields. Rani [20] found that PsyCap has a positive influence on the psychological well-being of unemployed youth. Many studies have also demonstrated the decisive role of PsyCap in helping workers fight negative symptoms associated with mental health [21]. Conversely, a study in the general population demonstrated that low PsyCap has negative impact on one's mental health [22]. Meanwhile, losing a job or experiencing a layoff is a traumatic experience, since it can not only disrupt the individual's sources of income, but it can also lead to a sudden loss of social status and changes in family relationship dynamics [23,24]. This could mean that people with low PsyCap who are unemployed may even suffer more serious mental health problems compared to the general population.

While previous studies have examined the mental health of unemployed individuals, there is a lack of evidence in the post-COVID-19 context and amid global turmoil. To fill the research gap, it is necessary to explore the relationship between mental health problems and PsyCap. Furthermore, academic interest in PsyCap topics has steadily increased over the past decade [25], confirming that PsyCap is making a significant contribution to the scientific foundation in diverse fields. Thus, this study will continue to emphasize the importance of PsyCap in the turbulent post-COVID-19 era by focusing on the context of layoffs and involuntary unemployment.

Additionally, according to World Health Organization statistics since the COVID-19 pandemic outbreak in Vietnam, the prevalence of common mental health problems is 14.9%, which corresponds to nearly 15 million people [26]. However, we have not found studies on the mental health of unemployed people in the post-COVID-19 context in Southeast Asian countries. Therefore, investigating mental health in the post-COVID-19 era would contribute to theory and provide more practical information.

Self-esteem is defined as one's positive or negative attitude toward the self [27]. Linn et al. [28] found that men with high self-esteem experienced less stress from job loss than men with low self-esteem. High self-esteem could also serve as a protective factor against the mental health effects of job loss because people with higher self-esteem engage in more intensive job-search efforts and possess a greater ability to emotionally distance themselves from the distress of involuntary job loss [29]. As a result, laid-off employees with high self-esteem may be more likely to find a new job within three months of losing their previous job [29]. Scholars have investigated self-esteem in different roles to determine its impact on mental health factors. Recent studies have shown that self-esteem can moderate the relation between mental health difficulties and related factors [30,31]. Therefore, self-esteem may play a moderating role in the relation between PsyCap and mental health problems.

Drawing from the literature, this study aims to investigate the level of depression, anxiety, and stress among employees experiencing job loss. The study also sought to examine the buffering effect of PsyCap on mental health problems, and the role of self-esteem in the relation between PsyCap and mental health problems of laid-off employees in Vietnam. In summary, we hypothesize the following:

H1: **PsyCap significantly predicts mental health problems among unemployed people in Vietnam.**

H2: **Self-esteem has a significant negative association with mental health problems and moderates the relation between PsyCap and mental health problems among unemployed people in Vietnam.**

## 2. Materials and methods

### 2.1. Study design

This study followed the quantitative method and a cross-sectional design

### 2.2. Participants

Participants were laid-off office employees at companies across a variety of industries, including technology, finance, consumer services, and infrastructure services.

### 2.3. Sample procedures

The researcher used an online questionnaire created by Google Survey posted on Facebook and LinkedIn platforms from May 25, 2023, to August 30, 2023, along with a letter calling for

sharing to laid-off individuals to participate to receive gifts. The letter of questionnaire fully explained the research objectives, ethics and instructions to the respondents. In addition, to ensure high response rate and reliability, the researcher carefully selected community support groups for laid-off people to deliver the questionnaire. Face-to-face questionnaires were sent to participants through human resources departments of companies in a variety of different fields.

## 2.4. Sample size

The sample size was calculated by G-Power software (version 3.1.9.7) based on a power of 0.95, an alpha of 0.05, and an effect size of 0.15 for linear multiple regression. The necessary minimum sample size for the current study was 166.

## 2.5. Instruments

**Psychological capital.** This was measured using the 12-item Psychological Capital Questionnaire - Short form (PCQ) [32], which was standardized in Vietnamese by Ha and Trung [33]. It consists of four subscales: hope (4 items), resilience (3), optimism (2), and self-efficacy (3). All items are rated on a six-point Likert scale from 1 (strongly disagree) to 6 (strongly agree). Each of the four PCQ subscale scores is calculated by taking the mean (average) of all items in the scale. The overall PsyCap score is calculated by taking the mean of all PCQ items, with scores ranging from 12 (the lowest possible score) to 72 (the highest possible score). The internal consistency of the scale was 0.92.

**Self-esteem.** This was measured using the 10-item Rosenberg Self-Esteem Scale (RSES) [27]. All items are rated on a four-point Likert scale from 1 (strongly disagree) to 4 (strongly agree). A total score of 10 to 40 is calculated by summing the scores of the ten items, with a higher total score indicates a higher sense of self-esteem. In which, scoring for items 2, 5, 6, 8, and 9 will be reversed. The Rosenberg Self-Esteem Scale was administered to Vietnamese secondary school students, demonstrating excellent reliability and validity [34]. The internal consistency of the scale was 0.9.

**Mental health problems.** These were measured using the 21-item Depression Anxiety Stress Scales - Short Form (DASS-21) [35], which has been standardized in Vietnamese by Le et al. [36]. It is composed of three subscales with seven items each: depression (DE), anxiety (AN), and stress (ST). All items are rated on a three-point Likert scale from 0 (Did not apply to me at all) to 3 (Applied to me very much or most of the time). Total scores are multiplied by 2 to reflect the original 42-item scale. Higher scores indicate greater levels of depression, anxiety, and stress. The scores range from 0 to 21 for each subscale and from 0 to 63 for the total scale. The DASS-21 provides standardized cut-off scores to categorize the severity of symptoms for each subscale. For depression: normal (0–9), mild (10–13), moderate (14–20), severe (21–27), and extremely severe (28 or higher). For anxiety: normal (0–7), mild (8–9), moderate (10–14), severe (15–19), and extremely severe (20 or higher). For stress: normal (0–14), mild (15–18), moderate (19–25), severe (26–33), and extremely severe (34 or higher). The internal consistencies of the three subscales were 0.76 (depression), 079 (anxiety), and 0.91 (stress).

**Suicidal ideation.** This was measured using item 9 of the Patient Health Questionnaire-9 (PHQ-9) [37], which states "Thoughts that you would be better off dead or of hurting yourself in some way." This has been standardized in Vietnamese by Nguyen et al. [38]. This measure was selected because the item has demonstrated a high degree of reliability in predicting suicidal ideation across different community contexts [39, 40]. This item is rated on a three-point Likert scale from 0 (not at all) to 3 (nearly every day). Higher scores indicate high levels of suicidal ideation. The reliability found in this study was 0.89.

## 2.6. Data analysis

For each study sample, we imputed missing data using the predictive mean matching (PMM) method [41, 42]. Analysis showed that the current study had no missing data. We used SPSS 25.0 software (SPSS Inc., Chicago, IL) to calculate descriptive statistics of the main variables (i.e., PsyCap, mental health problems, and self-esteem), Pearson correlations, and reliability coefficients. Variables were checked for distribution before analysis, skewness values between -1 and +1 and kurtosis values between -2 and +2 are typically considered acceptable for normality. Values outside of these ranges suggest deviations from a normal distribution.

Finally, we performed hierarchical regression analysis and used the SPSS PROCESS macro, Model 1 (version 3.0), to test the moderating function of self-esteem between PsyCap and mental health problems. PROCESS Model 1 was chosen because it is specifically designed to estimate moderation effects when a single moderator is present, aligning with the requirements of our hypothesis [43].

## 2.7. Ethics approval and consent to participate

The study was conducted according to the guidelines of the Declaration of Helsinki and approved by the Ethics Committee of the University of Social Sciences and Humanities, Vietnam National University, Ho Chi Minh City (ID: ERB 03-23). All participants provided written informed consent before completing the online questionnaire. Participants were informed that the study was voluntary, that they could withdraw at any time, and that their responses would remain anonymous.

## 3. Result

### 3.1. Sociodemographic characteristics of the study population and descriptive analysis of the variables

Table 1 presents descriptive statistics on the study variables for the 468 participants, who ranged in age from 22 to 49 years (mean age of 28.14 ± 3.84 years). Gender distribution included 50.4% male, 41.5% female, and 8.1% identifying as 'Other'. In terms of marital status, 59.4% were single, 35.3% married, and 5.3% divorced. Education levels varied, with the majority holding an undergraduate degree (85.7%), followed by 9.8% with a postgraduate degree and 4.5% with a high school education. When comparing mean scores based on demographic variables, no significant differences in mental health problems were observed across gender and age groups. However, among marital status and education groups, the mean scores for mental health problems (depression, anxiety, stress, and suicidal ideation) were higher in the postgraduate education and divorced categories compared to other groups.

Table 2 presents the average scores for depression, anxiety, and stress categorized according to the DASS-21 cut-off points [35]. Overall, the percentage of laid-off employees experiencing severe or extremely severe symptoms of depression, anxiety, and stress is remarkably high. Specifically, 54.5%, 50.9%, and 38.9% of unemployed people have "severe" to "extremely severe" symptoms of depression, anxiety, and stress, respectively.

### 3.2. Descriptive statistics and Pearson correlation

Table 3 presents the correlations between psychological capital (PsyCap), self-esteem, and mental health problems, including depression, anxiety, and stress, along with their means and standard deviations. The results show a significant positive correlation between PsyCap

**Table 1. Sociodemographic characteristics of post-COVID unemployment study in Vietnam.**

| Variable | N (%) | Depression (mean ± SD) | Anxiety (mean ± SD) | Stress (mean ± SD) | Suicidal (mean ± SD) |
|---|---|---|---|---|---|
| **Gender** | | | | | |
| Male | 236 (50.4%) | 11.19 ± 5.01 | 7.46 ± 4.37 | 11.41 ± 3.78 | .15 ± .36 |
| Female | 194 (41.5%) | 10.60 ± 4.42 | 7.76 ± 3.97 | 11.06 ± 3.56 | .15 ± .35 |
| Others | 38 (8.1%) | 9.89 ± 5.87 | 5.89 ± 4.79 | 10.84 ± 4.46 | .16 ± .37 |
| **Marital status** | | | | | |
| Single | 278 (59.4%) | 9.57 ± 4.70 | 6.63 ± 4.07 | 10.48 ± 3.71 | .12 ± .32 |
| Married | 165 (35.3%) | 12.32 ± 4.55 | 8.39 ± 4.11 | 12.08 ± 3.53 | .21 ± .42 |
| Divorce | 25 (5.3%) | 15.16 ± 3.11 | 10.56 ± 4.85 | 13.76 ± 3.29 | .16 ± .37 |
| **Education** | | | | | |
| High school | 21 (4.5%) | 6.14 ± 4.36 | 4.00 ± 3.28 | 8.00 ± 3.47 | .00 ± .00 |
| Undergraduate | 401 (85.7%) | 11.00 ± 4.59 | 7.52 ± 4.12 | 11.35 ± 3.51 | .14 ± .35 |
| Postgraduate | 46 (9.8%) | 11.54 ± 6.16 | 8.52 ± 5.13 | 11.50 ± 5.08 | .28 ± .45 |
| **Age** | | | | | |
| Under 30 years old | 380 (81.2%) | 10.66 ± 4.68 | 7.41 ± 4.06 | 11.07 ± 3.56 | .16 ± .37 |
| Over 30 years old | 88 (18.8%) | 11.59 ± 5.54 | 7.67 ± 5.08 | 11.85 ± 4.45 | .10 ± .30 |
| **Total** | 468 (100%) | 10.84 ± 4.86 | 7.46 ± 4.27 | 11.22 ± 3.75 | 0.15 ± .36 |

**Table 2. Levels of stress, anxiety, and depression among post- COVID unemployed people in Vietnam.**

| Level | Depression N (%) | Anxiety N (%) | Stress N (%) |
|---|---|---|---|
| Normal | 60 (12.8) | 108 (23.1%) | 75 (16.0%) |
| Mild | 20 (4.3%) | 25 (5.3%) | 71 (15.2%) |
| Moderate | 133 (28.4%) | 97 (20.7%) | 140 (29.9%) |
| Severe | 93 (19.9%) | 80 (17.1%) | 151 (32.3%) |
| Extremely Severe | 162 (34.6%) | 158 (33.8%) | 31 (6.6%) |
| Total | 468 (100%) | 468 (100%) | 468 (100%) |

and self-esteem (r = .81, p < .01). The results also indicate that PsyCap had significant negative correlations with depression (r = -.67, p < .01), anxiety (r = -.55, p < .01), and stress (r = -.57, p < .01).

## 3.3. Moderating effect analyses

Interaction effects were visualized by mean-centering the predictor variables and calculating the predicted means of the dependent variable for three levels of the predictors: one standard deviation below the mean, at the mean, and one standard deviation above the mean on the centered scales. Moreover, control variables are important in a moderator model because they support account for extraneous factors that might influence the dependent variable, ensuring a more accurate estimation of the relationships among the key variables of interest [43]. Therefore, we use demographic factors as control variables for all moderator models. In addition, the moderation model was implemented through the SPSS PROCESS macro with Model 1, and the moderating effect was determined based on 5,000 bootstrap samples in generating 95% bias-corrected bootstrap confidence intervals.

**Table 3. Correlations, means, and standard deviations among variables.**

| Measure | M | SD | Skewness | Kurtosis | PSY | SELF | DE | AN | ST |
|---|---|---|---|---|---|---|---|---|---|
| PSY | 3.48 | 0.76 | 0.56 | 0.89 | – | | | | |
| SELF | 24.70 | 5.40 | 0.62 | -0.39 | .81** | – | | | |
| DE | 10.84 | 4.86 | -0.39 | -0.57 | -.67** | -.70** | – | | |
| AN | 7.46 | 4.27 | 0.10 | -0.99 | -.55** | -.54** | .82** | – | |
| ST | 11.22 | 3.75 | -0.25 | -0.32 | -.57** | -.56** | .82** | .80** | – |
| SI | 0.15 | 0.36 | 0.69 | -0.73 | -.37** | -.34** | .32** | .37** | .27** |

Note: PSY = PsyCap, SELF = self-esteem, DE = depression, AN = anxiety, ST = stress, SI = suicidal ideation.

** $p < 0.01$.

### 3.4. Model 1: Stress as a dependent variable

Table 4 presents the moderation analysis results. For all moderation analysis, demographic variables were controlled. When stress is entered as the outcome, Table 4 showed that PsyCap (β = -.37, p < 0.01) and self-esteem negatively predicted stress (β = -.24, p < 0.01). Specifically, these two variables were found to explain 38% of variance in level of stress ($R^2$ = .38, p < 0.01). The interaction (PsyCap × self-esteem) also negatively predicted stress (β = -.10, p < 0.05). Overall, these results suggest that self-esteem can moderate the relation between PsyCap and stress among unemployed people.

To examine how self-esteem moderates the relation between PsyCap and stress, simple slopes were plotted for values of high self-esteem (1 SD above the mean) and low self-esteem (1 SD below the mean), as shown in Fig 1. The results indicated that, for the group with low self-esteem, a significant negative relation was observed between PsyCap and stress (B = -1.06, SE = 0.37, t = -2.91, p < 0.05, 95% CIs: -1.78, -0.35). This relation was also significant for the group with high self-esteem (B = -2.16, SE = 0.32, t = -6.66, p < 0.01, 95% CIs: -2.80, -1.52).

### 3.4. Model 2: Anxiety as a dependent variable

Table 4 showed that PsyCap negatively predicted anxiety (β = -.33, p < 0.01). Self-esteem also negatively predicted anxiety (β = -.24, p < 0.01). PsyCap and self-esteem were found to explain 36% variance in level of anxiety ($R^2$ = .36, p < 0.01). The interaction (PsyCap × self-esteem) did not negatively predict anxiety (p > 0.05). Overall, these results indicate that self-esteem cannot moderate the relation between PsyCap and anxiety among unemployed people.

### 3.5. Model 3: Depression as a dependent variable

The results (Table 4) showed that PsyCap negatively predicted depression (β = -.31, p < 0.01). Self-esteem negatively predicted depression as well (β = -.41, p < 0.01). PsyCap and self-esteem were found to explain 57% of variance in level of depression ($R^2$ = .57, p < 0.01). The interaction (PsyCap × self-esteem) also negatively predicted depression (β = -.10, p < 0.05). Overall, these results suggest that self-esteem can moderate the relation between PsyCap and depression among unemployed people.

To examine how self-esteem moderates the relation between PsyCap and depression, simple slopes were plotted for values of high self-esteem (1 SD above the mean) and low self-esteem (1 SD below the mean), as shown in Fig 2. The results indicated that, for the low-self-esteem group, a significant negative relation was found between PsyCap and depression (B = -1.25, SE = 0.40, t = -3.15, p < 0.05, 95% CIs: -2.03, -0.47). Also, this relation was significant for the high-self-esteem group (B = -2.32, SE = 0.35, t = -6.57, p < 0.01, 95% CIs: -3.02, -1.63).

**Table 4. Moderation analyses of self-esteem between mental health problems and PsyCap.**

**Stress (Fig 1)**

| | | Crude model | | | | | | | Adjust model | | | | | | |
|---|---|---|---|---|---|---|---|---|---|---|---|---|---|---|---|
| | Variables | $R^2$ | p | SE | β | t | p | 95%CI | $R^2$ | p | SE | β | t | p | 95%CI |
| Step 1 | PSY | .36 | p < 0.01 | .31 | -.35 | -5.48 | p < 0.01 | (-2.30, -1.08) | .38 | p < 0.01 | .31 | -.37 | -5.72 | p < 0.01 | (-2.41, -1.18) |
| | SELF | | | .04 | -.28 | -4.47 | p < 0.01 | (-0.28, -0.11) | | | .04 | -.24 | -3.72 | p < 0.01 | (-0.25, -0.08) |
| Step 2 | PSY | .37 | p < 0.01 | .31 | -1.56 | -5.04 | p < 0.01 | (-2.17, -0.95) | .40 | p < 0.01 | .31 | -1.61 | -5.15 | p < 0.01 | (-2.23, -1.00) |
| | SELF | | | .04 | -.18 | -4.13 | p < 0.01 | (-0.27, -0.10) | | | .04 | -.15 | -3.33 | p < 0.01 | (-0.23, -0.06) |
| | PSY × SELF | | | .03 | -.08 | -2.81 | p < 0.01 | (-0.13, -0.02) | | | .03 | -.10 | -3.75 | p < 0.01 | (-0.16, -0.05) |

**Anxiety**

| | | Crude model | | | | | | | Adjust model | | | | | | |
|---|---|---|---|---|---|---|---|---|---|---|---|---|---|---|---|
| | Variables | $R^2$ | p | SE | β | t | p | 95%CI | $R^2$ | p | SE | β | t | p | 95%CI |
| Step 1 | PSY | .33 | p < 0.01 | .36 | -.33 | -5.14 | p < 0.01 | (-2.55, -1.14) | .36 | p < 0.01 | .36 | -.33 | -5.03 | p < 0.01 | (-2.54, -1.11) |
| | SELF | | | .05 | -.27 | -4.21 | p < 0.01 | (-0.32, -0.12) | | | .05 | -.24 | -3.69 | p < 0.01 | (-0.29, -0.09) |
| Step 2 | PSY | .33 | .26 | .36 | -1.90 | -5.25 | p < 0.01 | (-2.61, -1.19) | .36 | .74 | .37 | -1.85 | -5.02 | p < 0.01 | (-2.57, -1.12) |
| | SELF | | | .05 | -.22 | -4.32 | p < 0.01 | (-0.32, -0.12) | | | .05 | -.19 | -3.70 | p < 0.01 | (-0.29, -0.09) |
| | PSY × SELF | | | .03 | .04 | 1.12 | .26 | (-0.03, 0.10) | | | .03 | .01 | .34 | .74 | (-0.05, 0.07) |

**Depression (Fig 2)**

| | | Crude model | | | | | | | Adjust model | | | | | | |
|---|---|---|---|---|---|---|---|---|---|---|---|---|---|---|---|
| | Variables | $R^2$ | p | SE | β | t | p | 95%CI | $R^2$ | p | SE | β | t | p | 95%CI |
| Step 1 | PSY | .52 | p < 0.01 | .35 | -.29 | -5.36 | p < 0.01 | (-2.53, -1.17) | .57 | p < 0.01 | .34 | -.31 | -5.77 | p < 0.01 | (-2.63, -1.29) |
| | SELF | | | .05 | -.46 | -8.50 | p < 0.01 | (-0.51, -0.32) | | | .05 | -.41 | -7.61 | p < 0.01 | (-0.46, -0.27) |
| Step 2 | PSY | .52 | .04 | .35 | -1.74 | -5.01 | p < 0.01 | (-2.43, -1.06) | .58 | p < 0.01 | .34 | -1.79 | -5.24 | p < 0.01 | (-2.46, -1.12) |
| | SELF | | | .05 | -.41 | -8.22 | p < 0.01 | (-0.50, -0.31) | | | .05 | -.35 | -7.27 | p < 0.01 | (-0.44, -0.26) |
| | PSY × SELF | | | .03 | -.06 | -2.04 | .04 | (-0.12, -0.00) | | | .03 | -.10 | -3.35 | p < 0.01 | (-0.16, -0.04) |

**Suicidal Ideation (Fig 3)**

| | | Crude model | | | | | | | Adjust model | | | | | | |
|---|---|---|---|---|---|---|---|---|---|---|---|---|---|---|---|
| | Variables | $R^2$ | p | SE | β | t | p | 95%CI | $R^2$ | p | SE | β | t | p | 95%CI |
| Step 1 | PSY | .14 | p < 0.01 | .03 | -.28 | -3.83 | p < 0.01 | (-0.20, -0.06) | .15 | p < 0.01 | .04 | -.25 | -3.30 | p < 0.01 | (-0.19, -0.05) |
| | SELF | | | .01 | -.11 | -1.54 | .12 | (-0.02, 0.00) | | | .01 | -.13 | -1.75 | .08 | (-0.02, 0.00) |
| Step 2 | PSY | .18 | p < 0.01 | .03 | -.16 | -4.56 | p < 0.01 | (-0.22, -0.09) | .19 | p < 0.01 | .04 | -.14 | -3.96 | p < 0.01 | (-0.21, -0.07) |
| | SELF | | | .01 | -.01 | -2.12 | .04 | (-0.02, -0.00) | | | .01 | -.01 | -2.24 | .03 | (-0.02, -0.00) |
| | PSY × SELF | | | .00 | .01 | 4.61 | p < 0.01 | (0.01, 0.02) | | | .00 | .01 | 4.20 | p < 0.01 | (0.01, 0.02) |

*Note*: PSY = PsyCap, SELF = self-esteem, SE = standard error, β = standardized coefficients. Adjusted models were control for age, gender, marital status, education level.

## 3.6. Model 4: Suicidal ideation as a dependent variable

The results (Table 4) showed that PsyCap negatively predicted suicidal ideation (β = -.25, p < 0.01). Self-esteem did not predict suicidal ideation (p > 0.05). PsyCap and self-esteem were found to explain 15% of the variance in suicidal ideation ($R^2$ = .15, p < .01). However, the interaction (PsyCap × self-esteem) positively predicted suicidal ideation (β = .01, p < 0.05). Overall, these results suggest that low self-esteem can moderate the relation between PsyCap and depression among unemployed people.

To examine how self-esteem moderates the relation between PsyCap and suicidal ideation, simple slopes were plotted for values of high self-esteem (1 SD above the mean) and low self-esteem (1 SD below the mean), as shown in Fig 3. The results indicated that, for the group with low self-esteem, a significant negative relation was observed between PsyCap and suicidal

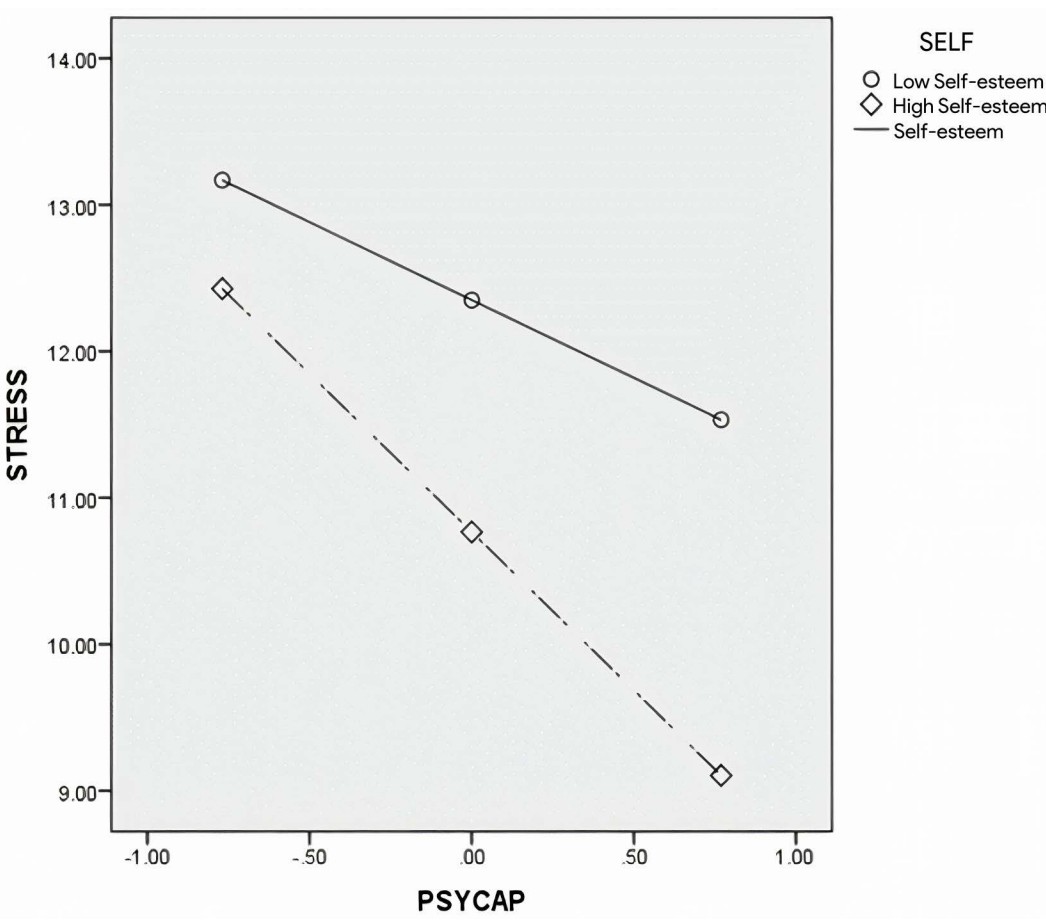

**Fig 1. The plot of simple slopes for self-esteem in the relation between stress and PsyCap.**

ideation (B = -.21, SE = 0.04, t = -5.08, p < 0.01, 95% CIs: -0.29, -0.13), but this relation was not significant for the group with high self-esteem (p > 0.05).

## 4. Discussion

To the best of our knowledge, this study was the first to investigate the relation between PsyCap and mental health issues while also examining the moderating role of self-esteem in such a relation. It yielded several notable findings.

First, the exploratory results showed that recently unemployed individuals in Vietnam suffered from high psychological distress. Specifically, based on the DASS-21 cut-off scores, more than 50% of the participants experienced severe to extremely severe depression and anxiety, and approximately 40% suffered from severe to extremely severe stress. Our study was similar to a Swedish study that suggested a higher level of mental disorders experienced by recently unemployed people as opposed to the employed sample [44]. Our results generally suggest that more attention be paid to the mental health problems of the unemployed.

Our results also demonstrated a significant relation between PsyCap and the dependent variables; that is, PsyCap was negatively associated with depression, anxiety, stress, and suicidal ideation. Accordingly, our findings support H1. These results were also consistent with studies conducted among nurses [45], international students [46], and mental health professionals [47], which found that low PsyCap is a significant risk factor for depression, stress, and anxiety.

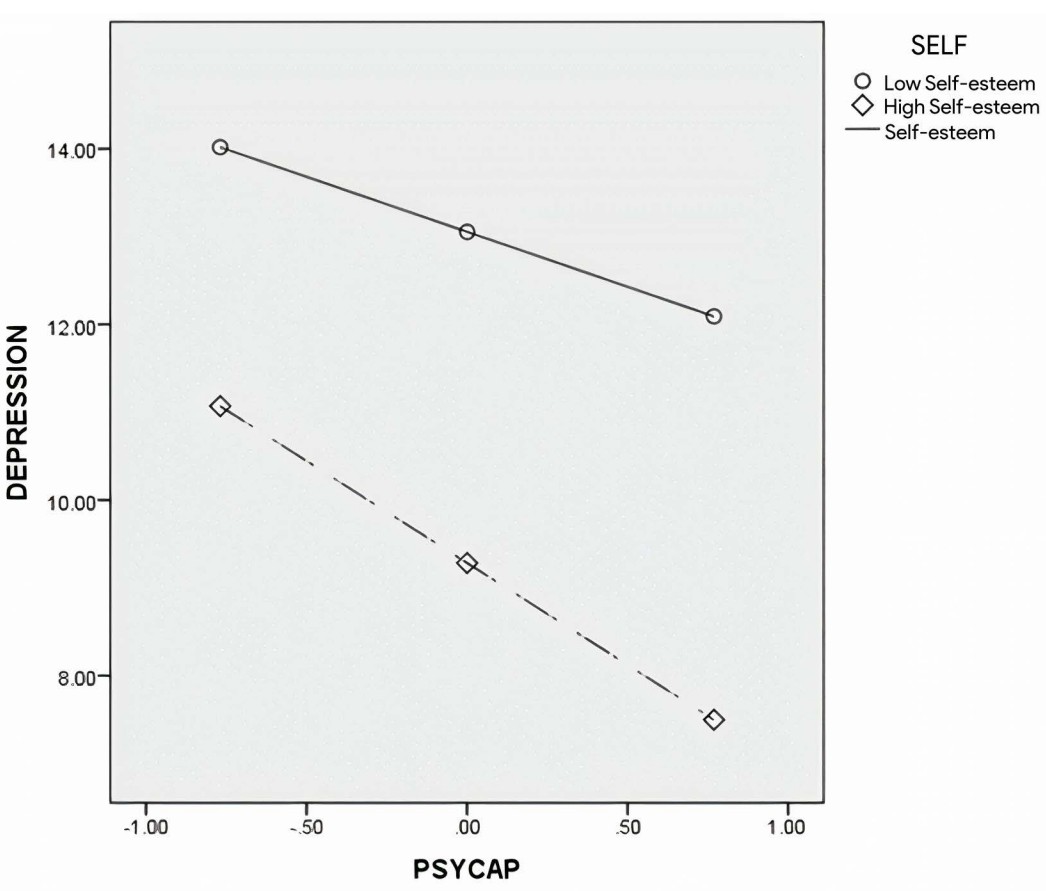

**Fig 2. The plot of simple slopes for self-esteem in the relation between depression and PsyCap.**

Furthermore, psychological distress has also been identified as an important predictor of suicidal ideation among Chinese employees [48], with individuals with higher levels of PsyCap are less likely to be affected by mental health problems. PsyCap could also foster a positive perspective of negative events, leading to improved mental well-being and mitigating the perception of job insecurity among employees [21]. Additionally, Chen and Lim [49] also suggested that unemployed individuals with high PsyCap tend to have faith in their skills and abilities as well as hold a positive view of future outcomes, thus showing more perseverance in their job-search instead of falling into despair.

H2, which states that self-esteem plays a moderating role in the relation between PsyCap and mental health issues, was partially supported by the current findings. Moderation analysis results showed that self-esteem significantly moderated the relation between PsyCap and stress, depression, and suicidal ideation but not the one between PsyCap and anxiety. Simply put, a high level of self-esteem may act as a buffer between lower PsyCap and depression, stress, and suicidal ideation. These results were consistent with other reports on the moderating role of self-esteem in predicting mental health issues [30,50,51]. These results align closely with behavioral plasticity theory [52], which posits that self-esteem can moderate the relationship between environmental stressors (e.g., being unemployed) and adaptive behaviors. In other words, individuals with low self-esteem are more malleable due to external influences, while those with high self-esteem show more stable responses. The theory also proposes that self-esteem plays a buffering role by making individuals with high self-esteem less affected by stressors, as they are less influenced by environmental cues and have greater self-confidence, which enhances their

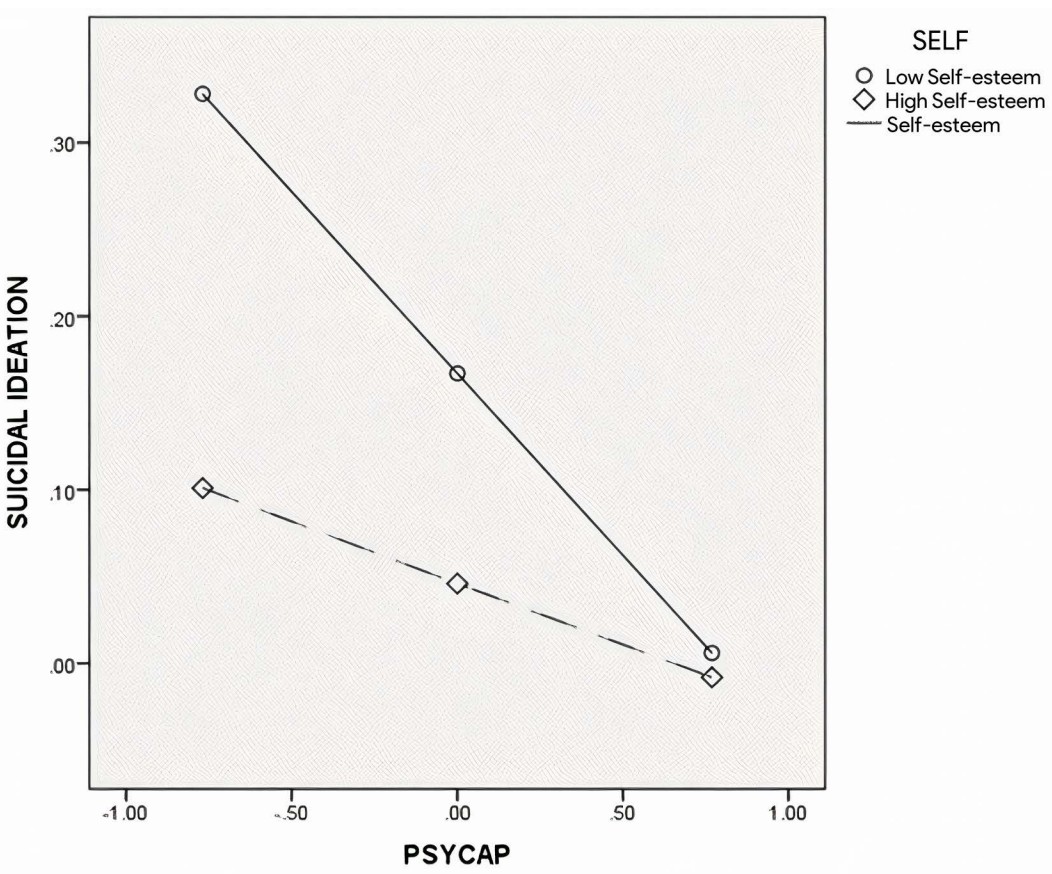

**Fig 3. The plot of simple slopes for self-esteem in the relation between suicidal ideation and PsyCap.**

ability to cope with challenges [53]. In the context of our study, it is possible that individuals with high self-esteem are less influenced by stressors related to unemployment, such as the negative emotional impact of job loss, because they rely more on their own self-worth rather than external validation. Additionally, their greater self-confidence and coping abilities may help them maintain psychological stability and resilience in the face of the challenges associated with unemployment. Our results also extend prior findings supporting behavioral plasticity theory, such as those reported by Mäkikangas and Kinnunen [54], who showed that self-esteem significantly moderated work stressors and well-being among employees.

Contrary to our hypothesis, self-esteem did not moderate the relation between PsyCap and anxiety. There may be several reasons for this finding. First, a meta-analysis indicated that while low self-esteem contributes to the development of depression and not vice versa, the link between low self-esteem and anxiety is symmetric reciprocal in nature in that low self-esteem influences anxiety as much as anxiety influences low self-esteem [55]. Thus, in the current study, the moderating effect of self-esteem on anxiety may be not as strong as that on depression, stress, and suicidal ideation. Additionally, general anxiety levels among the unemployed may remain unchanged even though they are suffering from high work-related anxiety. This theory is partially supported by Wang et al. [11], who found no changes in the prevalence of anxiety among the Canadian working population after an economic recession. Because the anxiety subscale in the DASS-21 only measures general anxiety and not work-related anxiety [56], the relation between the independent variable, moderator with anxiety might be weaker in comparison with other outcomes for the present sample.

### 4.1. Clinical implications

Our findings indicate that low PsyCap is a significant risk factor for mental health problems among the unemployed. Thus, enhancing one's PsyCap should be a main focus in improving their mental health during an economic crisis. This may be done through psychological capital intervention (PCI), a novel intervention program developed by Luthans et al. [57] and expanded upon by Dello Russo and Stoykova [58]. PCI is a microintervention that typically consists of a one- to four-hour group session designed to adopt various methods to improve all four PsyCap components among participants (i.e., efficacy, hope, resilience, and optimism) [57]. PCI has shown effectiveness in improving mental well-being [59] and reducing depression [60]. Accordingly, applying this program to unemployed individuals may also provide significant benefits in addressing mental health challenges during such crises. Additionally, previous intervention studies have demonstrated that PCI can reduce depression and improve job performance in employees even at follow-up periods, highlighting its sustained benefits [60, 61]. Given these findings, these interventions could also be integrated into employee mental health programs while proactively.

These findings also suggest that interventions for unemployed people incorporate self-esteem enhancement to foster a greater sense of mental well-being after a layoff. Indeed, interventions that include self-esteem enhancement have been suggested to be effective in reducing suicidal ideation [62]. To improve self-esteem, cognitive behavior therapy and compassion focused therapy have demonstrated efficacy in treating individuals with low self-esteem [63, 64].

### 4.2. Limitations

This study has several limitations that should be addressed in subsequent studies. First, the cross-sectional design limits the ability to draw causal conclusions between the study variables. Longitudinal studies would be more effective in establishing causal relationships and tracking changes over time. Second, the reliance on self-report instruments introduces the possibility of response bias, which may affect the validity of the results. This limitation could be mitigated in future investigations by incorporating multi-source data collection, such as using both self-report and observational measures, or adopting longitudinal designs that capture psychological outcomes over time. Additionally, the study used convenience sampling via LinkedIn and Facebook, which may introduce selection bias, as individuals who use these platforms might not represent the broader population, particularly in terms of socio-demographic factors or mental well-being. Therefore, subsequent studies should consider more diverse sampling methods to improve generalizability. Finally, the sample was limited to unemployed individuals seeking permanent employment, while previous studies have shown that individuals in temporary employment often experience higher psychological distress than those in permanent positions [65]. Further investigations could explore this distinction by comparing the mental well-being of unemployed individuals with temporary employment to those with permanent employment, offering a more comprehensive understanding of employment types and their effects on mental health.

## 5. Conclusions

The results showed extremely high rates of depression and anxiety among unemployed workers. This study also demonstrated that PsyCap is negatively associated with mental health problems including depression, anxiety, stress, and suicidal ideation. Furthermore, self-esteem can moderate the relation between PsyCap and depression, stress, and suicidal ideation but not anxiety. Overall, this study provided new perspectives regarding the unemployment situation of the labor market in Vietnam during the post-COVID-19 period. It has

also provided knowledge by highlighting the importance of PsyCap and self-esteem to help workers overcome mental health difficulties during crises. Considering this, companies and governmental organizations ought to implement policies that offer psychological support to jobless workers to promote economic growth, reduce social welfare burdens, and enhance social stability. Finally, employees can mitigate mental distress during challenging events such as unemployment, particularly in the post-COVID-19 era, by enhancing their psychological capital (PsyCap) and self-esteem.

## Supporting information

**S1 File. Dataset file.**
(XLSX)

## Acknowledgements

The authors would like to thank all the companies in the technology, financial services, consumer services, and infrastructure services sectors for their assistance in data collection. We offer special gratitude to the participants of the study for their support.

## Author contributions

**Conceptualization:** Cao Tran Thanh Trung, Nguyen Tan Dat.

**Data curation:** Cao Tran Thanh Trung.

**Formal analysis:** Cao Tran Thanh Trung, Nguyen Tan Dat.

**Investigation:** Cao Tran Thanh Trung.

**Methodology:** Cao Tran Thanh Trung.

**Project administration:** Cao Tran Thanh Trung.

**Resources:** Cao Tran Thanh Trung.

**Software:** Cao Tran Thanh Trung, Nguyen Tan Dat.

**Supervision:** Nguyen Tan Dat, Choon Jin Teh, Poh Kiong Tee.

**Validation:** Cao Tran Thanh Trung, Nguyen Tan Dat.

**Visualization:** Cao Tran Thanh Trung, Nguyen Tan Dat.

**Writing – original draft:** Cao Tran Thanh Trung.

**Writing – review & editing:** Nguyen Tan Dat, Choon Jin Teh, Poh Kiong Tee.

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
