## [Decision Letter · Decision Letter 0]

20 Dec 2024

PONE-D-24-49710Psychological capital and mental health problems among the unemployed in the post-COVID-19 era: Self- esteem as a moderatorPLOS ONE

Dear Dr. Tan Dat,

Thank you for submitting your manuscript to PLOS ONE. After careful consideration, we feel that it has merit but does not fully meet PLOS ONE’s publication criteria as it currently stands. Therefore, we invite you to submit a revised version of the manuscript that addresses the points raised during the review process.

We look forward to receiving your revised manuscript.

Kind regards,

Zypher Jude G. Regencia, Ph.D.

Academic Editor

PLOS ONE

Journal Requirements:

Reviewers' comments:

Reviewer's Responses to Questions

**Comments to the Author**

1. Is the manuscript technically sound, and do the data support the conclusions?

Reviewer #1: Partly

Reviewer #2: Yes

Reviewer #3: Yes

2. Has the statistical analysis been performed appropriately and rigorously? 

Reviewer #1: Yes

Reviewer #2: Yes

Reviewer #3: No

3. Have the authors made all data underlying the findings in their manuscript fully available?

Reviewer #1: Yes

Reviewer #2: Yes

Reviewer #3: Yes

4. Is the manuscript presented in an intelligible fashion and written in standard English?

Reviewer #1: Yes

Reviewer #2: Yes

Reviewer #3: Yes

5. Review Comments to the Author

Reviewer #1: This study investigates the impact of psychological capital on mental health issues among unemployed individuals in Vietnam, with a particular focus on the moderating role of self-esteem.

1. The introduction effectively sets the stage for the importance of the study, linking the COVID-19 pandemic to unemployment and subsequent mental health challenges. This context is highly relevant, and the mention of specific data. helps establish the significance of the issue. However, there is a slight imbalance between global events (bankruptcies, rising inflation) and the focus on the local context. A stronger emphasis on the unique impact in Vietnam would better align the introduction with the focus of the study.

2. While the introduction outlines the concept of PsyCap, further clarification on how it works as a buffer in the context of job loss would improve understanding. A sentence or two on how each component of PsyCap contributes to mitigating mental health issues would be helpful.

3. Some sentences have unnecessary complexity that could be simplified for clarity and to avoid redundancy. For example line 44, the phrase "encountering numerous new challenges" is a bit redundant.

4. For methods section, start with the study design, then describe participants, procedures, and finally, the instruments in a more organized and logical sequence.

5. line 115, the phrase "office employees" could be more specific. Are these workers from specific companies, or is the sample general across industries?

6. The study mentions using "Google Survey" and "social networking platforms such as LinkedIn and Facebook" to collect data. The inclusion of social media platforms can be a good strategy, but the specific method of recruitment (e.g., how the survey link was distributed via these platforms, how the participants were reached) should be further explained.

7. The use of the Rosenberg Self-Esteem Scale (RSES) is appropriate. However, the section could provide more context on how the Vietnamese version was adapted or validated for the population under study. Mention if the scale has been adapted or modified for the specific cultural context or population, and provide additional validation details if possible.

8. Some sentences are long and may confuse the reader. Breaking these down into simpler structures would improve readability. For example, the sentence in line 268: "Additionally, Chen and Lim [43] also suggested that unemployed individuals with high PsyCap tend to have faith in their skills and abilities as well as hold a positive view of future outcomes, thus showing more perseverance in their job-search instead of falling into despair."

9. While the discussion refers to behavioral plasticity theory and previous studies, the theoretical implications could be tied more strongly to the observed results, linking the moderation effects of self-esteem more explicitly to the theoretical framework.

10. he limitations section could be more specific. For example, addressing the self-report bias could be enhanced with a more detailed discussion on how future studies might mitigate such bias (e.g., through longitudinal studies or multi-source data collection).

Reviewer #2: The article can be accepted for publication. There are, however some issues that need to be clarified:

1. As for the clinical implications, the authors suggested PCI and self-esteem enhancement. Should these interventions be given to employees after a layoff, or should these interventions be included in the employee mental health programs while the employee is still employed?

2. Lines 328-330, what socio-economic development are you referring to here?

3. Lines 330-331, I wonder where in the paper was this explained to include this in the conclusion (especially on building resilience)? You should rephrase this to capture the essence of your whole paper.

Reviewer #3: The manuscript addresses an important topic: the psychological effects of unemployment in the post-COVID-19 context. By examining psychological capital (PsyCap) and self-esteem as protective factors for mental health, the study addresses a highly relevant area of research that has practical implications for mental health interventions and workforce policies. However, several aspects of the manuscript could be strengthened to improve its overall clarity, methodological rigor, and alignment with established reporting standards.

Specific Comments and Recommendations:

1. Introduction and Objectives

o Rephrase lines 105–106 to better align with the objectives of the study.

2. Methods

o Consider organizing the Methods section into distinct subsections, including:

Study design

Study population

Inclusion/exclusion criteria

Measures (e.g., primary and secondary outcomes, exposure variables, confounders, and effect modifiers)

Study procedures

Sample size considerations

Data analysis

o In the Measures subsection:

Provide details on how each variable included in the final analysis was treated (e.g., cut-off values used).

Include information on the minimum and maximum scores expected for each of the tools used to assess the outcome variables, giving readers more context.

o Address sample size considerations: While the sample size may be adequate for the study’s main objective, include details on how the sample size calculation was performed.

o Provide information on how missing or implausible values were handled in the analysis.

o Specify whether the distribution of the variables was checked prior to analysis.

3. Data Analysis

o Specify the type of regression analysis performed.

o Clearly explain why PROCESS Model 1 was chosen for moderation.

o Justify the decision to control demographic variables in all models.

o Indicate whether key assumptions (e.g., linearity, multicollinearity) were tested.

4. Tables and Figures

o Revise table and figure titles to explicitly reflect "what," "when," and "where," ensuring they align with the data presented.

o For Table 1, describe the 3rd to 6th columns in the text.

o Include 95% CI for regression results, and organize the table to clearly distinguish between crude and adjusted results.

o For very small p-values, avoid reporting them as 0.000; instead, use p < 0.01.

o Ensure information presented in the Results section (lines 164, 207–210, 226–228, 240) is reflected in the tables/figures.

5. Results

o Expand the text on the regression analysis results to include the magnitude of associations, providing a more comprehensive view of their strength and relevance.

o Highlight key findings instead of enumerating all the numbers in the text.

o Double-check figures mentioned on lines 213 and 218.

o Correct the term "independent" on line 191 to "dependent."

6. Discussion and Limitations

o Acknowledge the potential for selection bias due to convenience sampling via LinkedIn and Facebook in the Discussion section.

7. General Recommendations

o Improve the interpretation of results by emphasizing their implications while maintaining clarity.

o Ensure consistent formatting and adherence to reporting guidelines throughout the manuscript.

o Although ethics approval is mentioned, explicitly state how consent was obtained (e.g., checkbox or written agreement in the online form).

6. PLOS authors have the option to publish the peer review history of their article (what does this mean? ). If published, this will include your full peer review and any attached files.

**Do you want your identity to be public for this peer review?** For information about this choice, including consent withdrawal, please see our Privacy Policy .

Reviewer #1: No

Reviewer #2: No

Reviewer #3: **Yes: ** Olivia Sison

---

## [Author Response · Author response to Decision Letter 1]

22 Jan 2025

RE: Detailed responses for reviewers. Manuscript number PONE-D-24-49710

We would like to express our deep gratitude to the editor and the three reviewers for their thorough evaluation and constructive comments regarding our manuscript. Thanks to your comments, the article was improved significantly.

We are sending herewith our revised manuscript with some revision according to the comments of the three reviewers. Regarding the details, we have addressed the reviewers’ comments as follows:

REVIEWER #1:

1. The introduction effectively sets the stage for the importance of the study, linking the COVID-19 pandemic to unemployment and subsequent mental health challenges. This context is highly relevant, and the mention of specific data. helps establish the significance of the issue. However, there is a slight imbalance between global events (bankruptcies, rising inflation) and the focus on the local context. A stronger emphasis on the unique impact in Vietnam would better align the introduction with the focus of the study.

Author’s response: We have revised the manuscript as follows:

Line 44 – 50 (Old version)

As a result of this crisis, companies are encountering numerous new challenges, forcing them to implement mass layoffs or reduce their workforce to maintain operating costs [3, 4]. In 2023, more than 260,000 employees from over 1,100 technology companies experienced layoffs, an increase of nearly 58% compared with 2022 [5]. In Vietnam, the wave of layoffs started in early 2023 in diverse fields such as manufacturing, finance, real estate, and technology [6, 7]. Notably, the Ministry of Labor reported that 280,000 Vietnamese workers lost their jobs in the first 6 months of the year [8].

Revised:

As a result of this crisis, companies are encountering numerous new challenges, forcing them to implement mass layoffs or reduce their workforce to maintain operating costs [3, 4]. In 2023, more than 260,000 employees from over 1,100 technology companies experienced layoffs, an increase of nearly 58% compared with 2022 [5]. Although the global economy has been affected by the COVID-19 pandemic, Vietnam continues to be one of the few developing countries attracting strong Foreign Direct Investment (FDI) flows in the world [6]. However, over-reliance on FDI and exports has made the country's economic performance very vulnerable to external shocks [7]. This has led multinational corporations with branches in Vietnam to drastically cut headcount. Specifically, in Vietnam, the wave of layoffs started in early 2023 in diverse fields such as manufacturing, finance, real estate, and technology [8, 9]. Notably, the Ministry of Labor reported that 280,000 Vietnamese workers lost their jobs in the first 6 months of the year 2023 [10].

Newly added reference:

[6] Bui TT, Dang LH, Nguyen TMH, Nguyen GTH. Vietnam 2035 Report and Reform of Economic Institutions in 2016-2020. In: World Bank Group [Internet]. Jun 2021 [cited 1 Jan 2024]. Available: https://documents1.worldbank.org/curated/en/099111106102248474/pdf/P1647370c434f001b08ac408dd5dc320fbc.pdf

[7] Hiep LH. Vietnam’s Over-reliance on Exports and FDI. 2020 [cited 1 Jan 2024]. Available: https://www.iseas.edu.sg/wp-content/uploads/2020/09/ISEAS_Perspective_2020_96.pdf

2. While the introduction outlines the concept of PsyCap, further clarification on how it works as a buffer in the context of job loss would improve understanding. A sentence or two on how each component of PsyCap contributes to mitigating mental health issues would be helpful.

Author’s response: We have revised the manuscript as follows:

Line 62-70 (Old version)

Psychological capital (PsyCap) is considered a personal resource that is frequently observed among employees in various business environments [15, 16]. PsyCap is defined as “a state of positive psychological development of an individual”, and characterized by four components: hope, resilience, optimism, and self-efficacy [15]. Based on positive psychology constructs and empirical research, PsyCap refers to an individual’s ability to manage and react to difficult situations, facilitating the development of healthy individuals [15]. In other words, PsyCap, as highlighted by Luthans and Youssef-Morgan [15], is instrumental in preventing mental health issues by fostering adaptive coping strategies and empowering individuals to confront life's challenges with resilience and optimism.

Revised

Psychological capital (PsyCap) is considered a personal resource that is frequently observed among employees in various business environments [17, 18]. PsyCap is defined as “a state of positive psychological development of an individual”, and characterized by four components: hope, resilience, optimism, and self-efficacy [17]. Based on positive psychology constructs and empirical research, PsyCap refers to an individual’s ability to manage and react to difficult situations, facilitating the development of healthy individuals [17]. Each component of PsyCap plays a vital role in reducing mental health problems. As a synergistic effect, hope and optimism work together create a positive mindset, while self-efficacy and resilience strengthen an individual's belief in their ability to overcome hindrances [17, 19]. In other words, PsyCap, as highlighted by Luthans and Youssef-Morgan [17], is instrumental in preventing mental health issues by fostering adaptive coping strategies and empowering individuals to confront life's challenges with resilience and optimism.

Newly added reference:

[19] Newman A, Ucbasaran D, Zhu F, Hirst G. Psychological capital: A review and synthesis. J Organ Behav. 2014;35: S120–S138. doi: 10.1002/job.1916.

3. Some sentences have unnecessary complexity that could be simplified for clarity and to avoid redundancy. For example line 44, the phrase "encountering numerous new challenges" is a bit redundant.

Author’s response: We have revised the manuscript as follows:

Line 44-46 (Old version)

As a result of this crisis, companies are encountering numerous new challenges, forcing them to implement mass layoffs or reduce their workforce to maintain operating costs [3, 4]

Revised:

As a result of this crisis, companies are forcing them to implement mass layoffs or reduce their workforce to maintain operating costs [3, 4].

Line 95-98 (Old version)

High self-esteem could also serve as a protective factor against the mental health effects of job loss because people with higher self-esteem engage in more intensive job-search efforts and possess a greater ability to emotionally distance themselves from the distress of involuntary job loss [26].

Revised

High self-esteem could serve as a protective factor against the mental effects of job loss. People with higher self-esteem tend to engage in more intensive job-search efforts and are better able to emotionally distance themselves from the distress of involuntary job loss [26]

Line 264 – 267 (Old version):

Additionally, Chen and Lim [43] also suggested that unemployed individuals with high PsyCap tend to have faith in their skills and abilities as well as hold a positive view of future outcomes, thus showing more perseverance in their job-search instead of falling into despair

Revised:

Additionally, Chen and Lim [43] suggested that unemployed individuals with high PsyCap tend to have faith in their skills and abilities. This faith enables them to maintain a positive view of future outcomes and demonstrate greater perseverance in their job search, rather than succumbing to despair.

4. For methods section, start with the study design, then describe participants, procedures, and finally, the instruments in a more organized and logical sequence.

Author’s response: We have revised the manuscript as follows:

Revised:

Study design

This study followed the quantitative method and a cross-sectional design

Participants

Participants were laid-off office employees at companies across a variety of industries, including technology, finance, consumer services, and infrastructure services.

Procedures

The researcher used an online questionnaire created by Google Survey posted on Facebook and LinkedIn platforms from May 25, 2023 to August 30, 2023, along with a letter calling for sharing to laid-off individuals to participate to receive gifts. The letter of questionnaire fully explained the research objectives, ethics and instructions to the respondents. In addition, to ensure high response rate and reliability, the researcher carefully selected community support groups for laid-off people to deliver the questionnaire. Face-to-face questionnaires were sent to participants through human resources departments of companies in a variety of different fields.

5. line 115, the phrase "office employees" could be more specific. Are these workers from specific companies, or is the sample general across industries?

Author’s response: We have revised the manuscript as follows:

Line 115 (Old version): Participants were office employees in the fields of technology, finance, consumer services, and infrastructure services who have been laid off recently.

Revised:

Participants were laid-off office employees at companies across a variety of industries, including technology, finance, consumer services, and infrastructure services.

6. The study mentions using "Google Survey" and "social networking platforms such as LinkedIn and Facebook" to collect data. The inclusion of social media platforms can be a good strategy, but the specific method of recruitment (e.g., how the survey link was distributed via these platforms, how the participants were reached) should be further explained.

Author’s response: This point was addressed in Reviewer 2’s comment 4.

7. The use of the Rosenberg Self-Esteem Scale (RSES) is appropriate. However, the section could provide more context on how the Vietnamese version was adapted or validated for the population under study. Mention if the scale has been adapted or modified for the specific cultural context or population, and provide additional validation details if possible.

Author’s response: We have revised the manuscript as follows:

Revised

The Rosenberg Self-Esteem Scale was administered to Vietnamese secondary school students, demonstrating excellent reliability and validity [34].

Reference:

[34] Nguyen DT, Wright EP, Dedding C, Pham TT, Bunders J. Low Self-Esteem and Its Association With Anxiety, Depression, and Suicidal Ideation in Vietnamese Secondary School Students: A Cross-Sectional Study. Front Psychiatry. 2019;10. doi: 10.3389/fpsyt.2019.00698.

8. Some sentences are long and may confuse the reader. Breaking these down into simpler structures would improve readability. For example, the sentence in line 268: "Additionally, Chen and Lim [43] also suggested that unemployed individuals with high PsyCap tend to have faith in their skills and abilities as well as hold a positive view of future outcomes, thus showing more perseverance in their job-search instead of falling into despair."

Author’s response: This concern was addressed in Reviewer 1’s 3rd comment.

9. While the discussion refers to behavioral plasticity theory and previous studies, the theoretical implications could be tied more strongly to the observed results, linking the moderation effects of self-esteem more explicitly to the theoretical framework.

Author’s response: We have revised the manuscript as follows:

Line 268-282 (Old version)

H2, which states that self-esteem plays a moderating role in the relation between PsyCap and mental health issues, was partially supported by the current findings. Moderation analysis esults showed that self-esteem significantly moderated the relation between PsyCap and stress, depression, and suicidal ideation but not the one between PsyCap and anxiety. Simply put, a high level of self-esteem may act as a buffer between lower PsyCap and depression, stress, and suicidal ideation. These results were consistent with other reports on the moderating role of self-esteem in predicting mental health issues [27, 44, 45]. Our findings were also consistent with behavioral plasticity theory [46], which states that high-self-esteem individuals are less susceptible to environmental events (i.e., being unemployed) than low-self-esteem individuals. The plasticity hypothesis also proposes that high-self-esteem individuals are less affected by organizational events because they tend not to regard social cues and environmental stimuli as guides for their behavior like those with low self-esteem; thus, they are less likely to develop psychological symptoms in response to these environmental stressors. This theory is also supported by Mäkikangas and Kinnunen [47], who showed that self-esteem significantly moderated work stressors and well-being among employees.

Revised:

H2, which states that self-esteem plays a moderating role in the relation between PsyCap and mental health issues, was partially supported by the current findings. Moderation analysis results showed that self-esteem significantly moderated the relation between PsyCap and stress, depression, and suicidal ideation but not the one between PsyCap and anxiety. Simply put, a high level of self-esteem may act as a buffer between lower PsyCap and depression, stress, and suicidal ideation. These results were consistent with other reports on the moderating role of self-esteem in predicting mental health issues [30, 50, 51]. These results align closely with behavioral plasticity theory [52], which posits that self-esteem can moderate the relationship between environmental stressors (e.g., being unemployed) and adaptive behaviors. In other words, individuals with low self-esteem are more malleable due to external influences, while those with high self-esteem show more stable responses. The theory also proposes that self-esteem plays a buffering role by making individuals with high self-esteem less affected by stressors, as they are less influenced by environmental cues and have greater self-confidence, which enhances their ability to cope with challenges [53]. In the context of our study, it is possible that individuals with high self-esteem are less influenced by stressors related to unemployment, such as the negative emotional impact of job loss, because they rely more on their own self-worth rather than external validation. Additionally, their greater self-confidence and coping abilities may help them maintain psychological stability and resilience in the face of the challenges associated with unemployment. Our results also extend prior findings supporting behavioral plasticity theory, such as those reported by Mäkikangas and Kinnunen [54], who showed that self-esteem significantly moderated work stressors and well-being among employees.

New add reference:

[53] Saks AM, Ashforth BE. The role of dispositions, entry stressors, and behavioral plasticity theory in predicting newcomers’ adjustment to work. J Organ Behav. 2000;21: 43–62. doi: 10.1002/(sici)1099-1379(200002)21:1%3C43::aid-job985%3E3.0.co;2-w.

10. The limitations section could be more specific. For example, addressing the self-report bias could be enhanced with a more detailed discussion on how future studies might mitigate such bias (e.g., through longitudinal studies or multi-source data collection).

Author’s response: We have revised the manuscript as follows:

Line 312-319 (Old version)

This study has several limitations that must be addressed. First, it implemented a cross sectional study design, which makes it difficult to derive causal relations between study variables. Second, the study variables were measured using self-report instruments, possibly causing response bias and affecting the results. Finally, we only collected data from unemployed individuals with permanent employment. Bjarnason and Sigurdardottir [57] suggested that employees under temporary employment suffer from more severe psychological distress than those under permanent employment. Hence, future studies could offer a deeper understanding of types of employment by comparing mental well-being between unemployed individuals with temporary employment and those with permanent employment

Revised:

This study has several limitations that should be addressed in subsequent studies. First, the cross-sectional design limit

---

## [Editor Report · Decision Letter 1]

5 Feb 2025

Psychological capital and mental health problems among the unemployed in the post-COVID-19 era: Self- esteem as a moderator

PONE-D-24-49710R1

Dear Dr. Nguyen Tan Dat,

We’re pleased to inform you that your manuscript has been judged scientifically suitable for publication and will be formally accepted for publication once it meets all outstanding technical requirements.

Kind regards,

Zypher Jude G. Regencia, Ph.D.

Academic Editor

PLOS ONE
---

## [Editor Report · Acceptance letter]

PONE-D-24-49710R1

PLOS ONE

Dear Dr. Dat,

I'm pleased to inform you that your manuscript has been deemed suitable for publication in PLOS ONE. Congratulations! Your manuscript is now being handed over to our production team.

Kind regards,

on behalf of

Dr. Zypher Jude G. Regencia

Academic Editor

PLOS ONE
